

# Effects of wood chip amendments on the revegetation performance of plant species on eroded marly terrains in a Mediterranean mountainous climate (Southern Alps, France)

V. Breton[1], Y. Crosaz[2], F. Rey[1]

[1] Irstea, UR Ecosystèmes montagnards, BP 76, 38 402 Saint-Martin-d'Hères, France
Univ. Grenoble Alpes, 38402 Grenoble, France.
[2] Géophyte, 64 rue des Ecrins, 38530 Pontcharra, France

*Correspondence to:* V. Breton (vincent.breton@irstea.fr)

**Abstract.** The establishment of plant species can limit soil erosion dynamics in degraded lands. In marly areas in the Southern French Alps, both harsh water erosion and drought conditions in summer due to the Mediterranean mountainous climate prevent the natural implementation and regeneration of vegetation. Soil fertility improvement is sometimes necessary. With the purpose of revegetating such areas, we aimed to evaluate the effects of wood chip amendments on the revegetation performance of different native or sub-spontaneous plant species. We conducted two experiments on steep slopes over three growing seasons (2012–2014). The first consisted of planting seedlings (ten species), the second consisted of seeding (nine species including six used in the first experiment). First we noted that wood chips were able to remain in place even in steep slope conditions. The planting of seedlings showed both an impact of wood chip amendment and differences between species. A positive effect of wood chips was shown with overall improvement of plant survival (increasing by 11% on average, by up to 50% for some species). In the seeding experiment, no plants survived after three growing seasons. However, intermediate results for the 1st and 2nd years showed a positive effect of wood chips on seedling emergence: seeds of four species only sprouted on wood chips, and for the five other species the average emergence rate increased by 50%.

## 1 Introduction

Soil erosion and flooding degrade terrestrial ecosystems and affect vegetation development (Garcia-Ruiz et al., 2015). Considerable research has been conducted on the functioning and protection of soils from degradation, bringing the issues of soil conservation and the importance of soil ecosystem services to the forefront (Keesstra et al., 2012; Brevik et al., 2015). The protective role of vegetation against erosion processes is well known (Stokes et al., 2014). Given that a plant's functional traits determine the suitability of species to limit soil erosion (Stokes et al., 2009; Burylo et al., 2014a), the choice of plant species in bioengineering works is essential. Numerous methods commonly used for revegetation of degraded lands use living materials, which are able to resist hydrological and erosive forces on a sloping site. Three sources of propagules can be highlighted: i/ seeds that are possibly present on topsoil (soil seed bank) or brought by different techniques (seeding,



hydroseedling), ii/ plants, and iii/ cuttings. However, eroded areas often do not offer satisfactory conditions to support natural colonization or artificial revegetation, due to soil loss as well as low water and nutrient availability. Therefore, the success of bioengineering works and revegetation operations can depend on previous stages of soil fertility improvement (Donn et al., 2014; Young et al., 2015).

One way to improve soil conditions is to apply an organic amendment to the soil surface. Numerous studies have assessed the value of organic amendments on vegetation establishment and soil fertility, as for contaminated areas (Mahmoud and Abd El-Kader, 2015), post-mine soils (Eldridge et al., 2012; Benigno et al., 2013), semi-arid conditions (Jiménez et al., 2013; Tejada and Benítez, 2014) and eroded soils (Ojeda et al., 2003; Cohen-Fernández and Naeth, 2013; Prats et al., 2013; Hosseini Bai et al., 2014; Donn et al., 2014; Hueso-González et al., 2015). Others have also shown the direct effect of

organic mulch in reducing surface runoff (Moreno-Ramón et al., 2014; Cerda et al., 2015; Sadeghi et al., 2015). These organic amendments can have different forms: manure, green waste compost, straw, wood chips, etc. They can be incorporated into the soil or surface-applied. The type of amendment and the application requirements depend on site conditions (access, topography, soil). The effects, in particular when surface-applied (mulch), concern the soil water availability (van Donk et al., 2012). This practice conserves soil water by rainfall interception and reduction of soil

evaporation. It also reduces surface runoff and moderates soil temperature (Scopel et al., 2004).

Wood chips of small branches (Lemieux, 1988) are a form of organic amendment. Their use is developing, in particular on certain crops, even if validation provided by the scientific literature is incomplete (Barthès et al., 2010). Likewise, considering natural processes, the role played by woody litter on humus formation in forest ecosystems has not been sufficiently studied (Berg and McClaugherty, 2008). These authors admitted furthermore that white-rot fungi plays an

important role in woody material decomposition processes. The capacity of wood chips to improve the nutrient status of the soil depends on a quick and efficient stage of decomposition of organic matter and therefore on white-rot fungi presence and action. Therefore, Lemieux (1988) advised using low-diameter branches (less than 7 cm), which have a low carbone-to-nitrogen ratio and limit nitrogen removal used for wood degradation. Moreover, considering lignin resistance to degradation, it is recommended to favor deciduous trees rather than coniferous trees (Stevanovic, 2007).

In marly catchments of the French Southern Alps, extensive areas are subjected to intense hydric erosion, resulting in climatic events such as torrential floods specific to this mountainous Mediterranean climate: wetting-drying cycles, frost in winter, high-intensity rainfalls in summer, and consequently high sediment yields transported by floods at the exit of catchments (Yamakoshi et al., 2009). Experimental knowledge for ecological restoration of marly eroded lands is available through the voluminous research conducted in these areas (Rey, 2009; Burylo et al., 2012; Burylo et al., 2014a). These

studies have researched gully beds, where vegetation can counter erosive forces and trap sediment (Rey and Burylo, 2014; Rey and Labonne, 2015). It therefore appeared that vegetation could also be profitable in gully slopes where soil fertility is very low (soil loss, low soil moisture, and lack of an organic layer) and where natural vegetation is nearly nonexistent. The slopes are covered by a hard regolith layer, and vegetation cannot be established with the techniques currently used in gully beds, especially willow cuttings buried in soil. Planting and seedling methods for installing vegetation on slopes are



therefore required. Such methods have already been tested on numerous other eroded lands (Reubens et al., 2009; Bochet et al., 2010; Fernández et al., 2012; Commander et al., 2013; Lee et al., 2013) , but these studies do not apply to our climatic, topographic and pedological conditions, and the studied plant species are not appropriate or available. Therefore, the main issue for practitioners is to find the adequate solution to develop vegetation cover on gully slopes. The hydrological and

erosive forces are generally less marked than on gully floors but occasionally they can be strong. As on slopes and floors, the vegetation must allow both withstanding hydrological forces and trapping sediment as quickly as possible. The low fertility of the soil, which is in the form of a regolith layer composed of coarse particles embedded in a matrix of finer material, is evident on gully slopes: export of nutrients, very low water and nutrient availability, with no accumulation or mineralization of organic matter. For this reason, soil fertility improvement appears necessary.

We focused on crushed wood chips of small branches, considering that due to its size and form, this material may be better able to remain in place on slopes, compared to other forms of common amendment, finer and easily exported by runoff. This study aimed to test the effects of wood chip amendments on the revegetation performance of plant species on eroded marly terrains in a Mediterranean mountainous climate (Southern Alps, France). We hypothesized that in a context of water erosion and drought conditions of marly eroded land, the wood chip amendment, if able to remain in place, could facilitate plant

establishment. The first point was subjected to empirical observation. The second point was based on precise measurements on plants (counting, height, etc.). This study was based on two experiments, representative of two methods for installing vegetation: planting of seedlings and seedings (cuttings were dismissed owing to compactness and scarcely penetrable soil). They were carried out over three growing seasons (2012–2014). Different plant species were tested: 13 species – six were used in the two experiments – among those currently used and available in local tree nurseries, and a priori adapted to the

type of environment under study.

## 2 Methods

### 2.1 Site description

The site is located in the French Southern Alps (44°9′N, 6°21′E) near Digne in Alpes-de-Haute-Provence (Figure 1), in a badland area composed of gullies. The climate is mountainous and sub-Mediterranean, showing summer droughts with often

intense rainstorms. Over the three growing seasons of the experiments (2012–2014), the description of climatic parameters was based on data from the Sévigné meteorological station, 800–1000 m from both experimental sites. During this period, the mean annual total rainfall was 920 mm and the average annual temperature was 10.2°C. High climatic variations were observed both on the season scale (20°C differences in monthly temperatures between winter and summer), and on the year scale (long drought period in 2012). The 2012 summer drought was particularly severe: June and July showed a rainfall

deficit of 67 and 83%, respectively, compared to 30-year averages. Even more fluctuating are the intense rainfall events: rainstorms generally occur randomly and can be very heavy, especially in summer. Some rainfall intensities can reach 70



mm h$^{-1}$ lasting 1 h (Rey, 2009). Over the 3-year period, the most intense events (more than 50 mm h$^{-1}$ lasting at least 15 min) all occurred from the last 2 weeks of June to the end of August (once in 2012, three times in 2013, twice in 2014).

The soil composition depends on topographic conditions and alternates between a loose regolith layer made of disintegrated black marl fragment on the slopes and black marl sediment mainly on gully beds. The top layers are made of coarse marl fragments within a fine silty matrix and present low carbonate content, from 20% to 35%, with pH varying from 7.8 to 8.1 (Wijdenes and Ergenzinger, 1998). The slope gradient is relatively steep, reaching 75% in most cases on the steepest parts of the gullies. Spontaneous vegetation is present in a dispersed manner, mainly on the lower and higher parts of gullies, rarely on the slopes. The dominant tree species is *Pinus nigra*, and the shrub layer is mainly composed of *Juniperus communis*, *Hippophae rhamnoïdes* and *Buxus sempervirens*. Even more dispersed are *Ononis fructicosa*, *Lavandula officinalis* and *Robinia pseudo acacia*.

## 2.2 Plant and seed materials

We conducted two experiments to study two modes of revegetation over three growing seasons: 1) planting seedlings of ten plant species, hereafter designated as "plant experiment," and 2) seeding of nine plant species (Table 1), hereafter designated as "seed experiment." Six species were used in both experiments: *Acer campestre* L., *Alnus cordata* (Loisel.) Duby, *Buxus sempervirens* L., *Hippophae rhamnoïdes* L., *Juniperus communis* L., and *Lavandula officinalis* Chaix. Four species were tested only in the first experiment: *Quercus pubescens* Willd., *Pinus nigra* Arnold, *Robinia pseudo acacia* L., *Salix caprea* L., and three species only in the second experiment: *Dorycnium pentaphyllum* Scop., *Anthyllis vulneraria* L., *Ononis natrix* L. This set consists of ligneous and semi-ligneous species, mainly shrubs that commonly grow in eroded marly lands around the two experimental sites (except *Alnus cordata*). The choice of the species and the method of vegetation installation (seeds or young plants) depended on the plant material availabilities in the local tree nurseries and accorded with results of previous local studies (mentioned above, especially Burylo et al., 2014a).

Wood chips came from woody wastes of small branches after tree pruning in local public parks. It was mainly composed of poplars, lindens and plane trees. In the two cases, the wood chips were mixed and spread on the soil surface after seedling or planting operations, and formed a homogeneous 5-cm-thick layer. The seed densities depended on the species tested and were chosen according to the supplier's recommendations (Table 1). Because of the soil hardness, plant establishment required boring a 5-cm-diameter and 17-cm-deep hole with a drilling machine. This size corresponded to the size of the plant containers of all the species tested that were supplied by a local tree nursery.

## 2.3 Experimental design

For the two experiments, half of the surface area was covered with wood chips. The plant experiment consisted of five randomized completed blocks composed of two plots (control and wood chips). Each plot was composed of four replicates arranged in a row, and each row was composed of ten single trees of each species tested (Figures 2 and 3). Each plot covered 4 m². The seed experiment consisted of three randomized blocks divided into nine plots corresponding to the plant species.



Each plot was divided in two 1-m² half-plots corresponding to the soil treatment (control or wood chips). Each half-plot was 1 m × 1 m and divided into four 0.25 m² sub-plots (Figures 2 and 3). For the two experiments the blocks were generally placed in separate gullies. They were dispersed over an approximately 1-ha surface area, as far as possible on similar ecological conditions (slope, soil, hydric erosion).

## 2.4 Observations and measurements

The experiments were carried out over three growing seasons (2012–2014). Measurements and observations were conducted three times a year in order to divide the growing seasons: i/ at the end of May, ii/ at the end of July or the beginning of August, iii/ at the end of September or the beginning of October. This frequency allowed us to observe the rhythm of plant development during the growing seasons and to evaluate the effect of certain climatic parameters.

We quantified the seedling emergence of seeds by counting the number of seedlings of plant species sown on each sub-plot. The possibility of emergence of other plant species was also sought. For the plant experiment, seedlings were considered alive if living tissues in leaves, buds or stems were observed. We measured the plant height from the ground to the terminal bud of the tallest stem. The latter parameters were measured three times a year. The stem basal diameter was measured at the root–shoot junction only once a year at the end of the growing season. Compared to growth in height, we considered that growth in diameter was very low and did not need more than one measurement per year. For each measurement, an empirical observation of the state of the wood chip plots was made (export, covering, degradation).

## 2.5 Statistical analysis

Most of the data were non-normal, following different distributions and requiring nonparametric analysis. It was also necessary to involve random effects. We fitted generalized linear mixed models (GLMMs, Bolker et al., 2009) that can be performed on both normal and non-normal data and allowed us to analyze both fixed effects (soil treatment, species in the two experiments) and random effects (plot). Seedling emergence data were modeled using a GLMM with a Poisson distribution. Data on young plant mortality were analyzed with a binomial distribution. Young plant growth data (diameter, height) were analyzed using the linear mixed model (LMM) with a normal distribution. These analyses allowed us to research the effect of soil treatment that is composed of only two modalities (wood chips or control) with likelihood ratio tests. The species effect was also indicated but the differences between species cannot be known.

Seedling emergence data correspond to the maximum of seedlings out of the initial number of seeds in each plot, during the 1st year, the 2nd year, and all three years. Young plant survival data correspond to the number of living trees after the first, second and third growing seasons. The growth data were transformed to obtain relative data, with a difference between final measurement and initial measurement divided by initial measurement, in order to minimize the effect of initial plant size. All data were analyzed with the R statistical packages lme4.



## 3 Results

### 3.1 Wood chip observations

Over the 3 years, we clearly noted that wood chips remained in place and were hardly ever carried away by surface run-off. During the first few months (first and second measurements), white-rot fungi presence was observed for all the plots and
seemed to increase the cohesion of wood material. For the next 2 years, the white-rot fungi were not as visible. On the other hand, the material was partly covered by sediment.

### 3.2 Plant experiment

Considering plant survival, the experiment showed a significant effect of both species ($p= 0.003$) and soil amendment ($p<10^{-3}$, Table 2a). The rate of survival quickly decreased with *Alnus cordata*, *Lavandula officinalis* and *Salix caprea* to almost 0%
from the first growing season. They were removed for further analysis. The best survival rates were observed with *Acer campestre*, *Quercus pubescens*, *Pinus nigra*, *Robinia pseudo acacia* and *Buxus sempervirens* which exceeded 50%. The two other species, *Hippophae rhamnoïdes* and *Juniperus communis*, had intermediate results. A substantial positive effect of wood chip amendment on the survival rate was clearly observed with these two species: respectively, a 35 and 25% higher survival rate with wood chips. Most mortalities occurred during the summer period of the 1st year (Figures 4 and 5), which
corresponded to the most severe drought in comparison with the next 2 years.

Concerning the growth of young plants, significant effects were observed i/ for soil amendment with diameter and height measurements and ii/ for species with diameter only (Table 2a). There was a marked difference in the species ranking between diameter and height results (Figure 6). For instance, *Pinus nigra* showed higher relative growth in height than in diameter. On the contrary *Quercus pubescens* and *Hippophae rhamnoïdes* showed higher growth in diameter than in height.
Differences in plant architectural forms were obvious, with a shrubby form that showed no apical dominance (especially *Hippophae rhamnoides*) and an arborescent form (especially *Pinus nigra* and *Robinia pseudo acacia*). Moreover, some species (*Acer campestre*, *Quercus pubescens*) showed some dried apical stems, leading to problems assessing the initial plant growth and the comparison between species. Nevertheless, the results were very good for *Robinia pseudo acacia* and *Pinus nigra*, which doubled in height (and doubled in diameter for *Robinia pseudo acacia*) before the end of the 3rd year, and for
*Buxus sempervirens* to a lesser extent, which reached a 50% relative growth increase at that date. Conversely, we noted very low growth of *Quercus pubescens*, *Juniperus communis* and *Acer campestre*, which did not show real growth after three growing seasons.

### 3.3 Seed experiment

In all sub-plots we did not identify other species than the species sown. Analyses revealed that the effect of species and soil
amendment were significant in 2012 and 2013 (Table 2b). The date of emergence (Figure 7) separates two groups of species:
i/ *Alnus cordata*, *Hippophae rhamnoïdes*, *Dorycnium pentaphyllum*, *Anthyllis vulneraria*, *Ononis natrix* that sprouted mainly



in the 1st year and ii/ *Acer campestre*, *Buxus sempervirens*, *Juniperus communis*, *Lavandula officinalis* that sprouted mainly in the 2nd year. In both cases, no plants survived longer than one growing season. Three species sprouted new leaves for only one season (*Hippophae rhamnoïdes*, *Alnus cordata* and *Lavandula officinalis*); the others sprouted shoots over two seasons (Figure 7). We observed an obvious positive effect of wood chip amendment for most species during the first two

growing seasons. Some of them showed emergence capacity only with wood amendment: *Alnus cordata*, *Lavandula officinalis*, *Buxus sempervirens*, *Acer campestre* and *Juniperus communis*. The significance was clear even if it almost disappeared and no plants survived after the second growing season. Systematically the emergence rate decreased over the growing season. Whatever the year and the species, a higher rate always occurred in spring and it decreased during summer and autumn (Figure 7).

**4. Discussion**

Although based on empirical observations, we noted that wood chips remained in place during the observation period. On the basis of the meta-analysis of García-Ruiz et al. (2015), compared to numerous studies on erosion rates around the world, we can reasonably consider our experimental plots under very steep slope gradients and moderate mean annual precipitation. Moreover, local and particular events with very intense rainfall are possibly frequent in summer and were observed during

the 3 years. So, according to the climatic and topographic factors observed during the study period, wood chips seem able to remain in place under severe erosive constraints.

Wood chips offered the best results with seeds as well as young plants. The benefit was i/ evident as regards seedling emergence and plant survival and ii/ less efficient as regards plant growth. For the first point, these findings are in agreement with the reported performance of various mulches preserving soil water and consequently promoting seedling establishment

(Woods et al., 2012; Benigno et al., 2013; Hosseini Bai et al., 2014).

The positive effect on seedling emergence was obvious, although it did not last over 2 years. For four species, it only occurred with wood chip amendment: *Buxus sempervirens, Acer campestre, Juniperus communis* and *Lavandula officinalis*. For the five others, the positive effect of wood chips was often significant. We must also consider the possibility of the soil seed bank effect on seedling emergence. In each plot, we did not identify other species than the one sown. We cannot

dismiss natural emergence from the soil seed bank (instead of sowing); considering the very low spontaneous emergence rate in the vicinity of sown plots, this possibility appeared unlikely. These results are consistent with previous research on seedling emergence (Eldridge et al., 2012; Benigno et al., 2013) where soil treatment and tested plant species were different. Cerdà and García-Fayos (2002) noted the influence of the size and shape of seeds in their removal by water erosion. Our results are partly consistent, considering that the heaviest seeds (especially *Acer campestre*) were completely removed on untreated plots, but we did not have sufficient measurements on the seeds of the tested species to continue analysis of these

criteria. The same holds true for the effect of runoff and sediment yield on seed removal studied by Han et al. (2011). It is obvious that wood chip amendment improved seed emergence capacity. As we cannot determine whether the decisive role is



to avoid removal of seeds by water or to maintain soil moisture conditions, both cumulative effects must be considered. Other studies investigated the influence of organic amendment on the moisture of eroded soil (in particular Ojeda et al., 2003; Hueso-González et al., 2015), but the context of soils, types of amendment and slope gradients were significantly different, and the comparison is not relevant. According to Bochet et al. (2007), soil water availability after rainfalls

occurring during the germination period played a major role in seed germination. These authors and Crosaz (1995) also suggested that the species' ability to germinate under drought conditions could indicate a species' potential for success under semiarid conditions. In this way, considering the germinations during the strong drought period in 2012, we can assume some species' potentials, especially with *Hippophae rhamnoïdes, Dorycnium pentaphyllum, Anthyllis vulneraria, Ononis natrix* and finally *Buxus sempervirens* to a lesser extent, even if this ability did not last more than two seasons.

Regardless of soil treatment (wood chips or no wood chips), with an overall survival rate close to 50% (70% for some species) after three growing seasons, the planting of seedlings appeared to be an efficient revegetation technique for gully slopes. In comparison, the advantage of seeding was not proved. Reserves in ligneous tissues seem to ensure the best chances against dry summer conditions during the critical establishment period. Most plant mortalities occurred during the first summer, which was the driest of the 3 years, and occurred very little during the following summer droughts. Another

approach could be to associate seeding with drilling holes, as was experimented by Lee et al. (2013).

We also noted that differences in performance were generally greatest between species than between treatments (wood chips or control). The choice of species that were well adapted to drought conditions is essential and this study has provided additional data on that point. Several species showed the capacity to resist and remain in our experimental conditions (slopes, drought, flow erosion): *Robinia pseudo-acacia*, *Pinus nigra*, *Buxus sempervirens* hold both survival and growth capacities;

*Hippophae rhamnoïdes*, *Acer campestre*, *Quercus pubescens* maintained a satisfactory survival rate. To meet the objective of revegetation, these species must also show performance for erosion control. Thanks to the synthesis work of Burylo et al. (2014b), we have information on the response and effect traits of these species related to erosion dynamics in the same ecological conditions. The combined analysis of the two information sources allows us to improve the choice of appropriate plant species. Lastly, for the three species not tested in the young plant experiment (*Dorycnium pentaphyllum*, *Anthyllis*

*vulneraria*, *Ononis natrix),* the field capacity on gully slopes was not known and will require additional experimentation.

Numerous studies have emphasized the value of organic amendment for revegetation and erosion control (see the introduction). The novelty value of our results concerns the positive effect of wood chips for revegetation in a steep slope context. Bochet and García-Fayos (2004) underscored "… the difficulty of revegetating slopes with angles greater than 45° (100%), where the probability of seeds moving downhill is high." Other applications have to be examined in larger contexts

of restoration and bioengineering works. The question of wood chip supply must be also considered. Considering the requirements for wood chips in revegetation works (deciduous trees, low-diameter branches), the supply of such material can be difficult. The commercial competition with other wood uses (the wood-energy industry in particular) and the low production margins that cannot support transport costs, add further to these difficulties. As far as possible, solutions must be found in local production.



Finally, we must also investigate the contribution of these results for the bioengineering works in the context of degraded marly areas in the Southern French Alps. We already knew the possibilities and the limits of revegetation on gully beds (Rey, 2009). The main objectives of the two experiments were to assess revegetation techniques on gully slopes. The outcomes allow us to establish the requirements in those conditions such as planting seedlings of certain plant species (especially *Pinus nigra*, *Buxus sempervirens* and *Robinia pseudo-acacia*) that can be improved by a wood chip layer, in addition to willow cuttings and wooden sills on gully beds. Such operations will require new experiments to test the two bioengineering techniques on slopes and beds at the same time. Revegetation by seeding cannot be recommended on the basis of our results, even if the failure can be explained by very dry climatic conditions during the 1st year. Further experiments should be considered, in particular to determine whether seeding in autumn can limit the impact of a possible drought during the first growing season.

## 5. Conclusions

In the two experiments, the wood chip supply was distributed on steep slopes and in a context of water erosion. We showed that wood chips succeeded in remaining in place despite these erosive pressures and, at least during the first 2 years, have a positive effect on revegetation, mainly on plant survival and seed emergence capacities. This result is clearly confirmed in the seed experiment where, for several species, germination occurred only in the presence of wood chips during the second growing season, after intense rainfall events. The two experiments allowed us to test two methods of revegetation: seeding and planting of seedlings. We showed that revegetation on gully slopes is possible with the second method. The supply of wood chips can improve the establishment of young plants, limiting the mortality rate. The results with the seeding experiment were not as convincing. However, even if this study was limited in time, we noted seedling emergence capacity in association with wood chip amendment. All the possible applications of wood chips were not yet known, but we can consider that the gain in terms of plant survival and growth can be possibly decisive in revegetation work contexts, especially when extreme climatic events occur.

## Acknowledgements

Funding was received from the IngecoTech CNRS-Irstea program, and from Electricité de France (EDF), Agence de l'eau Rhône, Méditerranée et Corse, Région Provence-Alpes-Côte-d'Azur and the European Union (FEDER Programme "L'Europe s'engage en PACA avec le Fonds Européen de Développement Régional"). We thank P. Bourduge (Zygène) and P. Boutteaud (Vilmorin) for the seed supply and their technical advice. We also thank E. Bayle (ONF), N. Daumergue and P. Tardif for advice and assistance in setting up the experiment.

## Author contribution



VB, FR and YC conceived the research and the experimental design (YC for the seeding experiment, VB and FR for the planting experiment). VB (assisted by YC and FR) took the measurements, made observations and analyzed the data. The manuscript was written and edited by VB and FR.

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





**llustrations (tables and figures)**

**Table 1.** List of species tested in the two experiments and seed density (for species tested in the seed experiment)

| Plant species | | Seed density (number / m²) |
|---|---|---|
| **Tested in one experiment** | | |
| In the plant experiment | In the seed experiment | |
| *Quercus pubescens* Willd. (Que), | | |
| *Pinus nigra* Arnold (Pin), | *Dorycnium pentaphyllum* Scop. (Dor), | *4 000* |
| *Robinia pseudo acacia* L. (Rob), | *Anthyllis vulneraria* L. (Ant), | *8 000* |
| *Salix caprea* L. (Sal) | *Ononis natrix* L. (Ono) | *4 000* |
| **Tested in both experiments** | | |
| *Acer campestre* L. (Ace), | | *120* |
| *Alnus cordata* (Loisel.) Duby (Aln), | | *580* |
| *Buxus sempervirens* L. (Bux), | | *120* |
| *Hippophae rhamnoïdes* L. (Hip), | | *190* |
| *Juniperus communis* L. (Jun), | | *190* |
| *Lavandula officinalis* Chaix (Lav), | | *1 600* |

In parentheses: species abbreviations



**Table 2a.** Plant experiment: likelihood ratio tests for fixed effects (soil treatment in species) based on 1) generalized mixed-effects models (GLMM) for plant survival and on 2) linear mixed-effects model (LMM) for relative growth during the first three growing seasons.

| Effect | $\chi^2$ | $p$-value |
|---|---|---|
| Survival | | |
| Soil amendment | 8.60 | **0.003** |
| Species | 150.7 | **< 0.001** |

| Effect | $\chi^2$ | $p$-value |
|---|---|---|
| Relative growth in diameter | | |
| Soil amendment | 7.48 | **0.006** |
| Species | 124.98 | **< 0.001** |
| Relative growth in height | | |
| Soil amendment | 44.31 | **< 0.001** |
| Species | 0.09 | 0.764 |

$P$-values in bold indicate a significant effect (< 0.05)

**Table 2b.** Seed experiment: likelihood ratio tests for fixed effects (soil treatment and species) based on generalized mixed-effects models (GLMM) for maximum seedling emergence

| Effect | Year | $\chi^2$ | $p$-value |
|---|---|---|---|
| | 2012 | | |
| Soil amendment | | 22.72 | **< 0.001** |
| Species | | 427.25 | **< 0.001** |
| | 2013 | | |
| Soil amendment | | 309.14 | **< 0.001** |
| Species | | 379.73 | **< 0.001** |
| | overall | | |
| Soil amendment | | 183.77 | **< 0.001** |
| Species | | 164.59 | **< 0.001** |

$P$-values in bold indicate a significant effect (< 0.05)



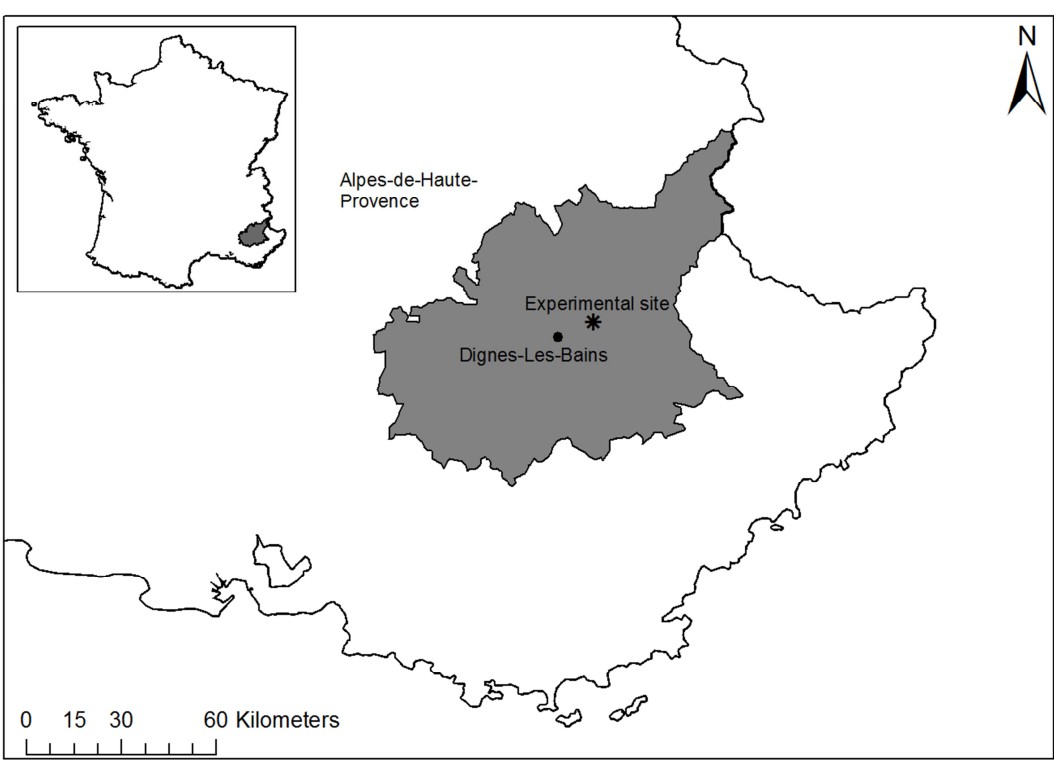

**Figure 1.** Location of the experimental site




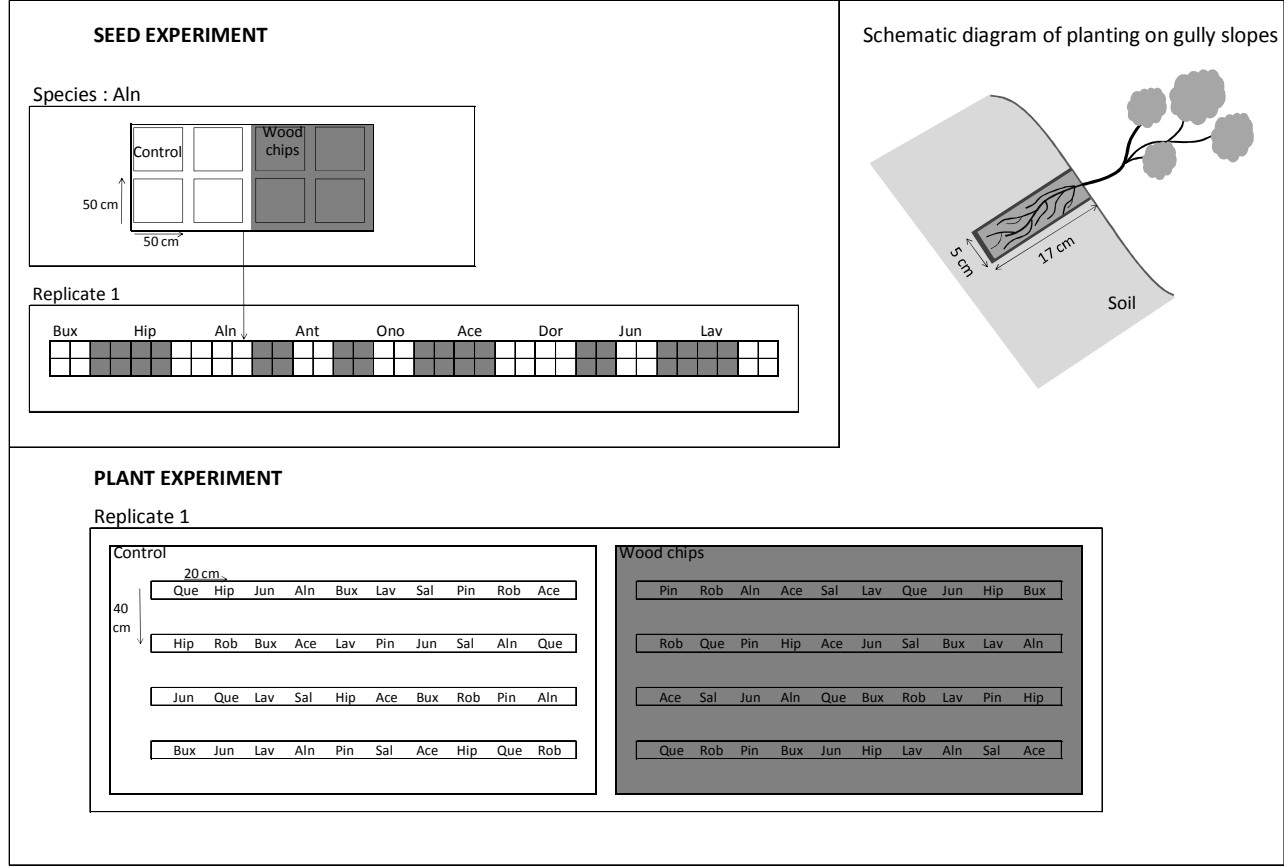

**Figure 2.** Experimental design of the two experiments (see species abbreviations in Table 1)

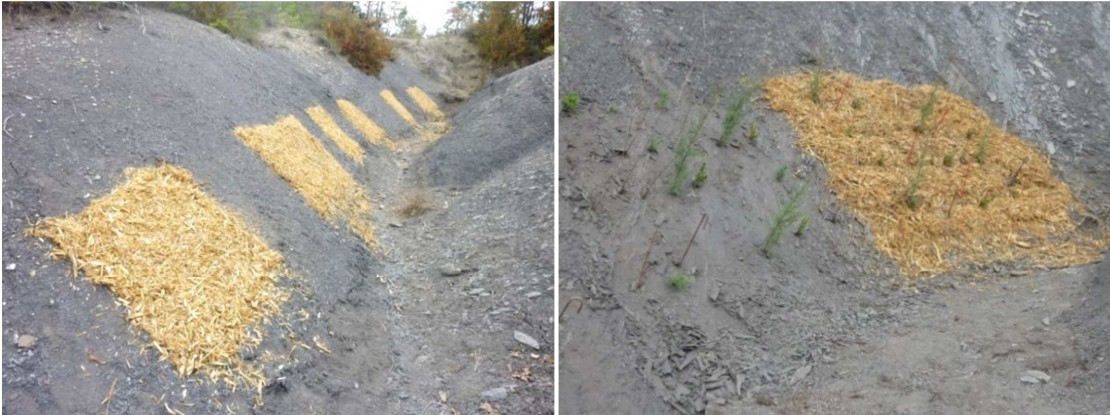

**Figure 3.** Pictures of the seed experiment set-up (left) and the plant experiment set-up (right)





**Figure 4.** Mortality rate of seedlings (bottom: number of dead trees observed from the next assessment) and climatic parameters during the three growing seasons (ombrothermic diagram with monthly rainfall (mm) = 2*monthly temperature (°C)). Black arrows indicate the observation dates. Mortality rates were calculated from dead plants per stage referred to the live plants at the previous stage.





**Figure 5**. Plant experiment: comparison of survival rates between "wood chips" and "control" modalities, for three growing seasons after planting (see species abbreviations in Table 1)





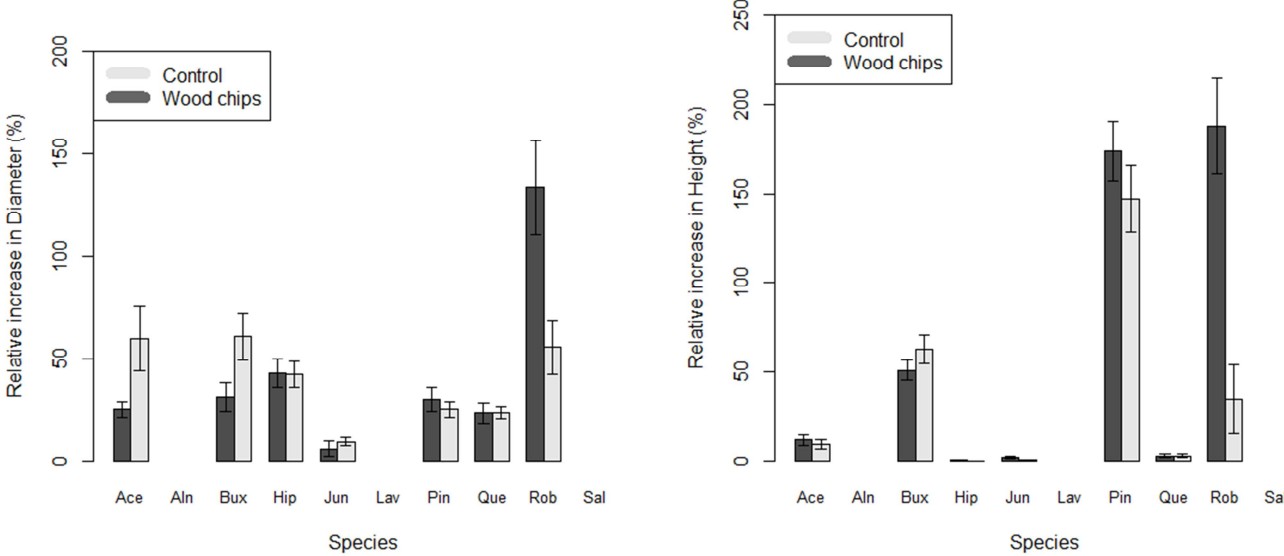

**Figure 6.** Relative increase in height (%, left); Relative increase in diameter (%, right) after three growing seasons. Bars indicate SE (see species abbreviations in Table 1)





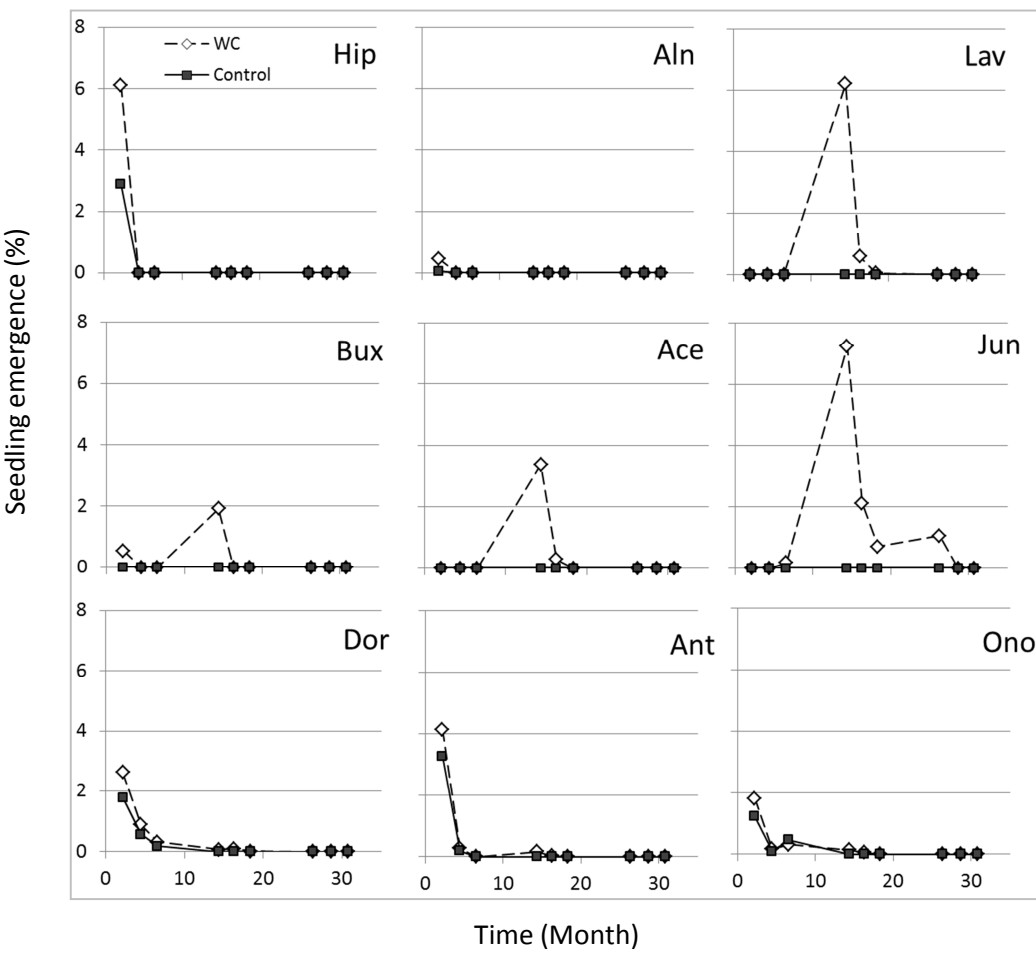

**Figure 7.** Effect of wood chips (WC) and comparison between species on seedling emergence for three growing seasons after seeding (see species abbreviations in Table 1)