# Peer review of "Effects of wood chip amendments on the revegetation performance of plant species on eroded marly terrains in a Mediterranean mountainous climate (Southern Alps, France)"

_Solid Earth, 2016_

## Referee Comment (RC1) · Anonymous Referee #1 · 8 Feb 2016

The paper is of good quality and shows interesting research how degraded land in gullies and badlands can be restored. However, it is important to keep in mind that the source of soil erosion in the world comes from agriculture land and this should be shown clearly in the paper.

You could get some information from these recent papers below that deal with this issue:

Debolini, M., Schoorl, J.M., Temme, A., Galli, M., Bonari, E.Changes in Agricultural Land Use Affecting Future Soil Redistribution Patterns: A Case Study in Southern

[Figure]

Tuscany (Italy)(2015) Land Degradation and Development, 26 (6), pp. 574-586 DOI: 10.1002/ldr.2217 Li Q. Y., Fang H. Y., Sun L. Y., Cai Q. G. Using the 137Cs technique to study the effect of soil redistribution on soil organic carbon and total nitrogen stocks in an agricultural catchment of Northeast China. (2014) Land Degradation and Development, 25 (4), pp. 350-359. Cited 2 times. DOI: 10. 1002/ldr. 2144 Cerdà, A., Flanagan, D.C., le Bissonnais, Y., Boardman, J.Soil erosion and agriculture(2009) Soil and Tillage Research, 106 (1), pp. 107-108. DOI: 10.1016/j.still.2009.10.006 Novara, A. Keesstra, S., Cerdà, A., Pereira, P., Gristina,L. 2016. Understanding the role of soil erosion on co2-c loss using 13c isotopic signatures in abandoned Mediterranean agricultural land. Science of The Total Environment, 550, 330-336, http://dx.doi.org/10.1016/j.scitotenv.2016.01.095. Ochoa, P.A., Fries, A., Mejía, D., Burneo, J.I., Ruíz-Sinoga, J.D., Cerdà, A. 2016. Effects of climate, land cover and topography on soil erosion risk in a semiarid basin of the Andes Catena, 140, 31-42.

And also you should highlight that mulch strategies contribute to reduce the soil losses also in agriculture land See here some examples Prosdocimi, M., Jordán, A., Tarolli, P., Keesstra, S., Novara, A., Cerdà, A. 2016. The immediate effectiveness of barley straw mulch in reducing soil erodibility and surface runoff generation in Mediterranean vineyards. Science of the Total Environment, 547, pp. 323-330. DOI: 10.1016/j.scitotenv.2015.12.076 Mwango, S.B., Msanya, B.M., Mtakwa, P.W., Kimaro, D.N., Deckers, J., Poesen, J.Effectiveness of mulching under miraba in controlling soil erosion, fertility restoration and crop yield in the usambara mountains, Tanzania. Land Degradation and Development, DOI: 10.1002/ldr.2332 Sadeghi S. H. R., Gholami L., Sharifi E., Khaledi Darvishan A., Homaee M. Scale effect on runoff and soil loss control using rice straw mulch under laboratory conditions. (2015) Solid Earth, 6 (1), pp. 1-8. Cited 6 times. DOI: 10. 5194/se-6-1-2015 Tejada, M., Benítez, C. Effects of crushed maize straw residues on soil biological properties. (2014) Land Degradation and Development, 25 (5), pp. 501-509.. DOI: http://dx.doi.org/10.1002/ldr.2316 Novara, A., Gristina, L., Saladino, S. S., Santoro, A., & Cerdà, A. (2011). Soil erosion assessment on tillage and alternative soil managements in a Sicilian vineyard. Soil and Tillage
[Figure]

Research, 117, 140-147.

Novara, A. Keesstra, S., Cerdà, A., Pereira, P., Gristina,L. 2016. Understanding the role of soil erosion on co2-c loss using 13c isotopic signatures in abandoned Mediterranean agricultural land. Science of The Total Environment, 550, 330-336, http://dx.doi.org/10.1016/j.scitotenv.2016.01.095.

─────────────────────

---

## Author Comment (AC1) · 25 Feb 2016

Thank you for considering our paper on effects of wood chips amendments in a context of ecological restoration. In your comments, you suggested to develop the issue of agriculture as source of erosion. We agree that the question is of importance, but our works do not really concern this issue. Therefore we prefer not developing it, especially for three reasons: - this question is not essential to the understanding of the article and the future prospects from this research do not concern agricultural issue. In fact, it was based on a local experiment and, as it is mentioned in the conclusion, the results

must be considered ". . . in revegetation work contexts . . ." (P8L11). - the subject of erosion in agricultural context is too vast and would make heavy the introduction. You suggested 9 references; a rapid search on Scopus website shows many other articles (62 documents with "erosion AND agriculture" in title, 37 documents with "erosion AND agriculture AND mulch" in keywords). Our paper includes at present 46 references, which is already quite a lot in our point of view. - the articles you suggested would increase the number of references from certain journals, which is not justified.

Moreover, two references that you suggested were already cited: Sadeghi S. H. R., Gholami L., Sharifi E., Khaledi Darvishan A., Homaee M. Scale effect on runoff and soil loss control using rice straw mulch under laboratory conditions. (2015) Solid Earth, 6 (1), pp. 1-8. Tejada, M., Benítez, C. Effects of crushed maize straw residues on soil biological properties. (2014) Land Degradation and Development, 25 (5), pp. 501-509.

---

## Referee Comment (RC2) · Anonymous Referee #2 · 7 Mar 2016

The paper of Breton et al. (se-2016-11) presented a study about the effects of wood mulching on the revegetation performance of plant species on marly terrains affected by gully erosion in Southern France. Generally speaking, the paper is interesting, clear and well written. In my opinion, it only requires few changes that would further improve the work. Information about the application rate of mulching is missing. You well described the composition of wood mulch but you did not report any information about how much mulch you had to apply to achieve a homogeneous 5-cm-thick layer. The application rate is a very important aspect for technical and management purposes because, together with the type of mulch, it drives the cost of the application. The

article is very interesting from a scientific point of view, but in my opinion, it should be enhanced with the above-mentioned information: i) application rate of mulching and ii) cost of the application rate: how much did you pay for getting the wood chips? In this way, the reader, who may also be a land manager, not only is aware of the effectiveness of wood mulching on the revegetation performance of plant species but also of the possible total cost. Follow some minor changes: 1) Please, in the whole text, refer to as "soil water erosion" rather than "hydric erosion". 2) Pag. 2, line 19: "play" rather than "plays" 3) Pag. 2, line 31: "Therefore, it…" rather than "It therefore…" 4) Pag. 3, line 26: "800-1000 m far from…"rather than "800-1000 m from…" 5) Pag. 4, line 1: "Over the 3-year period, all the most…" rather than "Over the 3-year period, the most". Then, you remove the "all" from the second line. 6) Paragraph 3.1: In my opinion, this paragraph should be removed because of its shortness. The information provided can be re-structured and reported in the other results. 7) Page. 7, line 12: "Garcia-Ruiz et al. (2015), who compared…" rather than "Garcia-Ruiz et al. (2015), compared…"

---

## Author Comment (AC2) · 18 Mar 2016

Thank you for your comments and suggestions.

Regarding the first point (application rate and cost), we gave additional information : - P4L25 (before "... Because of ..."): "The total volume of wood chips used to cover the different plots was 2 m3 (almost 1 m3 per experiment, corresponding to 500 m3/ha)." - P8L35: "This study has been carried out on small surface areas and with small quantities of mulch. Therefore it does not allow us to make a reliable analysis of the costs. In the experimental site, the supply of the wood chips cost between 30 and 50 euros/m3

(between 1,5 and 2,5 euros/m2). With large quantities, it is obvious that those prices must significantly decrease. Considering the total cost of the bioengineering project, the additional cost of wood chips (supply and application) represented a slight extent: 8-10 % for the plant experiment. As the seeds were free of charge, this rate cannot be estimated for the seed experiment."

For the second point (minor changes): we made corrections as you suggested.